# Analysis of Smart Home Technology Acceptance and Preference for Elderly in Dubai, UAE

Mohammad Arar, Chuloh Jung * , Jihad Awad and Afaq Hyder Chohan

Department of Architecture, College of Architecture, Art and Design, Ajman University, Ajman P.O. Box 346, United Arab Emirates; m.arar@ajman.ac.ae (M.A.); j.awad@ajman.ac.ae (J.A.); a.chohan@ajman.ac.ae (A.H.C.)
* Correspondence: c.jung@ajman.ac.ae; Tel.: +971-56-698-8344

**Abstract:** The elderly are the most predicted users for smart home technology in the United Arab Emirates and the population over 65 is expected to increase to 24.3% by 2030. Despite the rapid development of smart home technology, research has been mainly focused on technology development. To encourage conservative elderly users, however, smart home technology should be implemented for UAE elderly users to accept and integrate it into their daily lives. The objective of this paper is to analyze the preferences and needs of smart home technologies to understand the behaviors of UAE elderly users, and the factors affecting the acceptance of technology. As a methodology, a survey and interview were conducted for 110 people in their 40s and 60s and a total of 105 valid survey responses were collected and used as data for frequency, mean, cross-analysis, independent sample *t*-test, one-way variance analysis, and multiple regression analysis with IBM SPSS statistics 27. The results showed that 67.0% of UAE elderly users have chronic diseases such as high blood pressure (16.2%), heart disease (3.8%), diabetes (32.4%), or arthritis (10.5%). Therefore, smart home technology for health management is inevitable to improve overall lifestyles. It was statistically proven that UAE elderly users want automatic fall detection in the living room (39.0%) and bedroom (25.7%). Lifestyle monitoring in living room (44.7%) and bedroom (18.1%); the elderly preferred living room most for daily life assistance (36.2%), environmental control (50.5%), health and biometric monitoring (49.5%), and video conferencing (82.9%). In the case of sensors, elderly preferred the switch at the entrance (36.2%), and motion detecting sensors (42.9%), video cameras (56.2%), and voice recognition (50.5%) sensors in the living room. However, UAE elderly users do not think smart home technology can protect their privacy. It is found that age group and computer technology affinity are the most influential variables and UAE elderly users have an anxiety about technology, which influenced the acceptance of smart home technology.

**Keywords:** smart home technology; Dubai; United Arab Emirates; user preference; user acceptance





## 1. Introduction

Along with social change, the development of information and communication technology (ICT) has continuously changed the concept of housing [1–3]. With the recent advancement of new technologies such as Artificial Intelligence (AI) and the Internet of Things (IoT), the universalization of mobile phones and tablets, and the popularization of smart devices, interest and demand for a smart home is increasing further [4–7]. In particular, the elderly are the most predicted customers to generate demand for services of smart home technologies [8–11], and the implementation of many smart housing technologies (SHT) is targeting the elderly as the main target [12,13]. When deciding whether to stay in their home or move to an institutional facility, seniors aged 60 and over strongly hope to stay in their home for as long as possible [14–16]. Therefore, for the elderly with features such as functional and cognitive impairments, chronic diseases, reduction of social networks, and low physical activity, SHT improves the quality of life, reduces medical expenses, and allows them to live a more sustainable independent life at home [17,18].

Much effort has been made in developing technologies to support the health of the elderly, such as sensor-based networks for measuring physiological signals, monitoring activities, and detecting falls and roaming [19–21]. Despite the rapid development and growth of SHT over the past decade, smart home research so far has been mainly focused on technology development [22–24]. Numerous research projects have focused on implementing various prototypes and developing new technologies and capabilities such as sensors, algorithms, and intelligent devices [25,26]. However, recent research results show that there is considerable resistance from users to the increase in information technology in homes [27,28]. Residents lack a well-known concept of new and innovative information technology and a lack of understanding of technology-based solutions, and it is very difficult to introduce SHT, especially for conservative users such as the elderly [29–31].

For example, Home Telehealth Service, which combines ICT with the medical industry, is expected to save time and medical expenses for the elderly who need chronic diseases care, and to ensure a safe and independent life [32,33]. Users perceived remote medical services in their homes as potentially useful, but in reality, it was found that in many cases they refused or gave up access to personal health records, bio-signal measurement, and daily life monitoring through various sensing systems [34,35]. Users expressed their fear of using unfamiliar technology rather than the technical benefits provided by a customized residential environment that autonomously tracks and records even without adjustment [36,37]. In addition, concerns were raised that many areas were unnecessarily monitored, and privacy was not protected [38,39]. Shachar et al. (2020) pointed out that privacy is a major factor hindering the acceptance of technology, and that our society sometimes neglects or ignores personal information protection in order to emphasize the necessity of technology [40]. The United Arab Emirates, a regional benchmark for technological progress in Middle East, is among the leading markets for smart homes in the Middle East and the market in the Emirates is expected to grow at a CAGR of 14.8% during the period 2016–2022. However, challenge with the awareness of SHT still remains [41]. Like the case of other advanced countries, SHT is only effective if UAE elderly users accept it and integrate it into their daily lives. This suggests that it is very important not only to develop technology, but also to understand what technologies UAE elderly users prefer and whether they are ready to accept and use these technologies [42] (Figure 1).

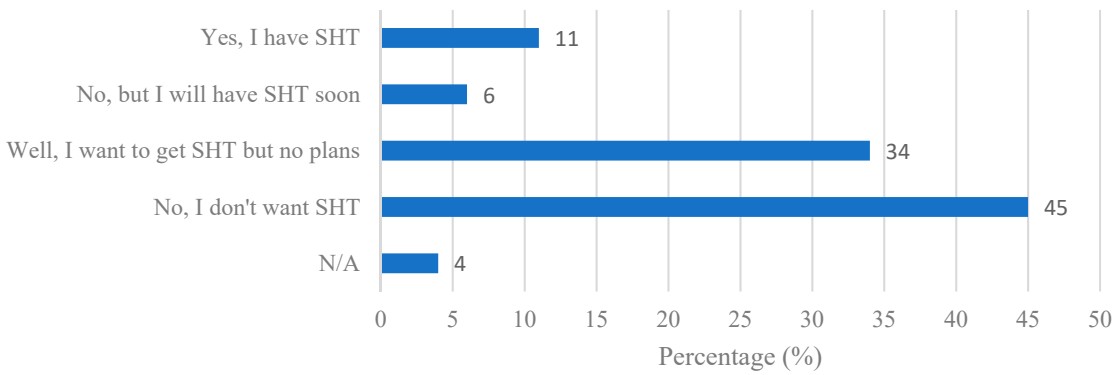

**Figure 1.** Age 50 and older who have SHT in UAE.

The Hartford Center for Mature Market Excellence and the MIT AgeLab performed the focus group interview with leading experts on housing, aging, and technology from the fields of occupational therapy, interior design, computer science, gerontology, and engineering, and a survey of homeowners with 25 smart home technologies [43]. The survey result identified the top ten smart home technologies that can make life easier, help with home maintenance, and enhance safety and security for homeowners over the age of 50. The top ten technologies included the following: (1) smart smoke and detector, (2) wireless doorbell cameras, (3) keyless entry, (4) smart lighting, (5) smart water valves, (6) smart home security systems, (7) smart outlets/plugs, (8) smart thermostats, (9) water

and/or mold monitoring sensors, and (10) smart window blinds [44]. They also identified a list of smart home technologies that benefit elderly with a health condition and help them to maintain their safety, independence, and well-being at home. The top five technologies on that list include: (1) telehealth systems, (2) medical management systems, (3) medication management systems, (4) smart fall detection systems, (5) smart beds/sleep sensors [45].

The purpose of this study is to investigate the UAE user's preference and perception of a wide range of technical solutions of SHT and identify the factors that influence the acceptance of SHT and the difference in technology acceptance behavior with TAM (Technology Acceptance Model) according to UAE elderly user characteristics. This study will serve as basic data to help design more efficient SHT hardware devices for the elderly.

## 2. Materials and Methods

A smart home is an Internet of Things (IoT) technology-based system that connects and remotely or automatically controls smart devices (home appliances, lighting, sensors, and security) indoors (Figure 1). In order to implement a smart home, communication technology such as a device to realize apps and services acting as a control hub, a cloud server, and a gateway to connect them are required. To link and control devices, a cloud server that is located in the cloud and provides functions to manage and link users/devices is required. There must be a gateway that connects the smart home device with the cloud server over the internet. Home appliances with Internet of Things (IoT) functions, sensors, lighting, and other devices that realize a smart home and apps that control devices must also be equipped. To implement a smart home, the app control function normally found on a smartphone was utilized as a hub. However, there is a recent movement to use apps on TVs with large screens as hubs. To implement a smart home, it is necessary to apply segmented communication technology. It is characterized by low power consumption so that it can be used in small and simple devices. Communication technology used in a smart home can be divided into low-power long-distance communication, low-power internet IP communication, and low-power RF communication. Low-power long-distance communications are typically used primarily for outdoor devices. It features low-speed, low-power, long-range technology. Communication technologies such as LoRa, LTE-NB, and SigFox are representative examples. Low-power IP communication is a protocol for communicating smart home gateways and cloud servers. CoAP, XMPP, and MQTT are used. Low-power RF communication is the technology used between the smart home gateway and the device. Bluetooth and ZigBee are representative technologies.

Smart home technology is designed to support and enable a safe, convenient, healthy, and independent life at home [46,47]. The most basic technical feature of a smart home is the ability to control facilities and devices automatically at home or from the outside [48,49]. With the development of home networking with the spread of high-speed Internet, users can easily control or monitor devices in their homes from outside [50]. In addition, it became possible to not only analyze the living patterns of residents, but also to communicate and collect information between smart devices and objects, and humans through new technologies such as Artificial Intelligence and Internet of Things, moving away from the concept of a home network that simply controls devices connected to the network [51,52]. Based on this, a technology that can provide customized services to residents by anticipating residents' wants and needs has been developed [53–55]. Many of the new technologies using various sensing systems, such as motion sensors and video cameras, are being developed to the point where they can support users without having to manually manipulate the device [56,57]. Examples of such new monitoring devices include a fall detection system, a lifestyle monitoring system, a physiological health monitoring system, and a lifestyle assistance system [58].

The biggest beneficiaries of smart housing technology can be the elderly and the physically disabled [59,60]. In particular, the healthcare sector is in the spotlight at the present time when the elderly population increases with the retirement period of the baby boomer generation [61,62]. Smart home health provides the next generation of medical

services for the elderly by allowing the family and caregivers to remotely monitor the elderly's health via technologies such as sensors and related algorithms. Ubiquitous computing applications can be used in predicting a fall based on a change in gait rather than recognizing and notifying it after a sudden accident occurs [63,64]. From cell phones to furniture, picture frames, kitchen utensils, toilets, and other intelligent devices in a variety of homes, they are used to motivate residents to manage their diet, take medicine, or keep exercising [65]. In addition, telemedicine technology that connects with clinicians to manage chronic diseases more comfortably at home than in expensive hospital environments or monitors physiological signals such as heart rate by wearing or attaching them to clothing or skin is becoming more and more common [45,66].

To convert a conventional home to smart home has become easier recently. Ready-made sensors, smart lighting bulbs, smart speaker, smart home security cameras, multi-room speaker systems, and smart smoke detectors can be integrated with a gateway (router) that helps them communicate with each other and with you via smartphone, smart tablet, or smart watch (Figure 2).

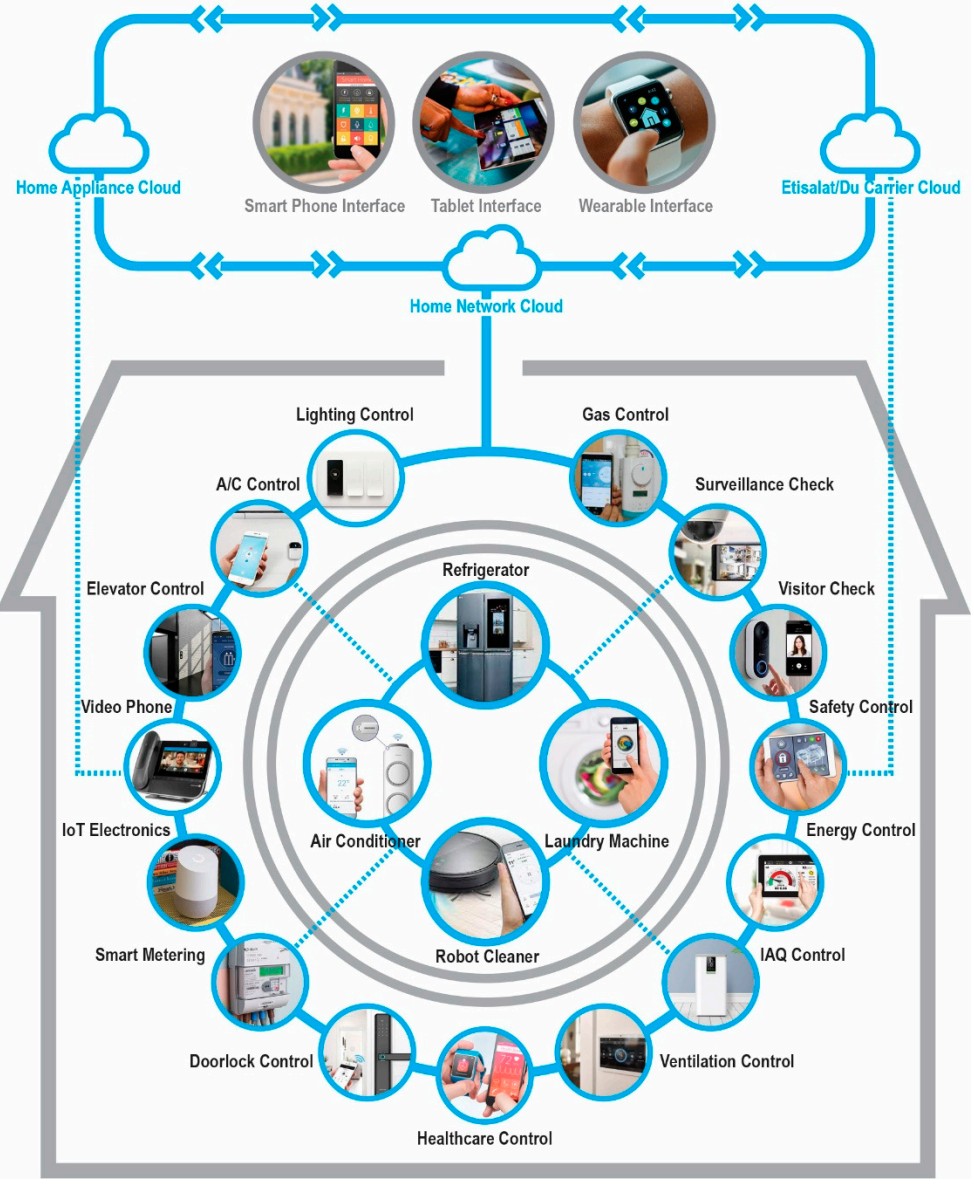

**Figure 2.** Smart Home with Integrated Smart Devices.

This study aims to analyze the behavior of UAE residents to adopting smart home technology based on the previously developed theory of technology acceptance. TAM (Technology Acceptance Model) is an information technology theory that models how a user accepts and uses a technology [67]. When a user encounters a new technology, it explains the acceptance intention and behavior of the technology, and the predictive factors that affect it [68]. This model suggests that the acceptability of information technology is determined by two main factors: perceived usefulness and perceived ease of use [69]. Holden and Karsh (2010) developed the TAM, which deals more specifically with predictions about the acceptability of information technology based on the theory of rational behavior [70]. Since then, TAM has been continuously researched and expanded. Among them, Unified Theory of Acceptance and Use of Technology (UTAUT) is a model developed by Venkatesh et al. (2016) and is widely used in various fields such as services, e-commerce, and electronic finance [71]. UTAUT aims to describe the user's intention and behavior intention to use the information technology and refers to the degree to which an individual intends to use the technology [72] and is influenced by four major constituents: performance expectancy, effort expectancy, social influence, and facilitating conditions [73]. The expected performance and expected level of effort are factors derived from the perceived usefulness and perceived ease of use of the first TAM model [74]. In addition to these influencing factors, a recent study on the acceptance of health technology introduced additional contextual predictors to the basic model of UTAUT to attempt a more accurate understanding of the user's technology acceptance [75]. Here, extended factors such as computer anxiety, security, and trust were used as important indicators for predicting user acceptance of technology [76]. Computer anxiety is an anxiety or emotional response that users feel when performing actions using a computer. The higher the computer anxiety, the more hesitant to use ICT-related products, and computer anxiety is an important variable predicting the acceptance of technology, especially in the elderly [77]. In addition, since smart home technology requires the transmission and management of personal health-related data, it is important to maintain security and build trust. Therefore, by introducing predictive factors for each context, the acceptance of user technology can be more accurately understood.

Based on the aforementioned predictive factors for UTAUT's technology acceptance and the conditions used in the acceptance of health technology, this study analyzes smart home technology's intention and expected performance, expected effort, social impact, facilitating conditions, anxiety about technology, the perceived security, and intention to use. Expected performance refers to the degree to which user believe that the use of technology will help user perform a specific activity, and expected effort refers to the degree of ease associated with using the system.

Social influence refers to the degree to which important people around me think they should use the system. An anxiety about technology is an emotional feeling such as anxiety that a user feels when using technology, and the perceived security refers to the degree to which personal information can be managed using information technology. To analyze each factor, measurements were developed (Table 1).

This study aims to analyze the preferences and needs of new smart home technologies in order to understand the intentions and behaviors of UAE elderly users, and also the factors affecting the acceptance of technology according to user characteristics are examined via survey (Appendix A). First, what types of smart home technologies do UAE elderly users prefer and where are their preferred locations in their homes? Second, what are the factors that hinder or promote the acceptance of smart home technology by UAE elderly users? Third, is there a difference in technology preference and acceptance according to UAE elderly users' characteristics such as age?

**Table 1.** Development of Measurement of Technology Acceptance Model (TAM).

| Category | Contents |
| --- | --- |
| Expected Performance | EP1. Smart Home Technology will improve the quality of my life<br>EP2. Smart Home Technology will be useful in everyday life<br>EP3. Smart Home Technology will improve my health |
| Expected Effort | EE1. Learning Smart Home Technology will be easy<br>EE2. The use of Smart Home Technology will be clear and easy to understand |
| Social Impact | SI1. People around me think I should use Smart Home Technology<br>SI2. People around me will support using Smart Home Technology |
| Facilitating Conditions | FC1. The initial environment to use Smart Home Technology needs to be set up<br>FC2. In case of a system failure, helps should be provided |
| Anxiety About Technology | AT1. Using Smart Home Technology can make me be anxious and uncomfortable<br>AT2. I am hesitant to use technology because I am afraid to make mistakes |
| Perceived Security | PS1. Using Smart Home Technology will not leak personal information<br>PS2. I think using Smart Home Technology is safe |
| Intention to Use | IU1. I will use Smart Home Technology in the future<br>IU2. I will always use Smart Home Technology in my daily life<br>IU3. I accept that technology changed my lifestyle |

According to the results of research related to elderly users' characteristics, the younger the age, the higher the education level, the higher the income, the more use of internet and smart phone [78]. In addition, the greater the desire to live in the same place, the greater the affinity with the technology, the greater the acceptance of smart home technology [79]. Shin et al. (2018) compared the differences between the elderly and baby boomers, emphasizing that opinions of each generation should be considered in order to grasp the differences between users' preference and acceptance of technology [80]. Younger generations have more exposure and experience to technology than older generations [81]. They are also more likely to live with their children in a home that uses computing technology even when they are older. Therefore, in this study, 40s and 60s were selected as survey targets in order to understand the differences and influencing factors on the adoption of technology by generation. The sampling process was conducted by a trained students to visit apartments located in JBL, JLT, and Business Bay in Dubai, and a random sampling was performed between 1 October 2020 and 2 December 2020. However, for the analysis of UAE user characteristics, a similar number of samples were needed in the case of gender and age, so the proportional allocation sampling method was applied to collect the samples. A questionnaire survey was planned by developing a questionnaire tool, but as a result of the preliminary survey, it was difficult to collect questionnaires in those aged in their 60s, so questionnaires for those in their 60s were separated from those in their 40s and completed. In this survey, people in their 60s conducted a questionnaire and interview at the same time. Over four weeks, a questionnaire survey was conducted for 55 people in their 40s, and a questionnaire and interview were conducted for 55 people in their 60s. In the survey, 5 of the subjects had some missing values and were treated as nonresponse. A total of 105 questionnaires were collected and used as data, and frequency, mean, cross-analysis, independent sample *t*-test, one-way variance analysis, and multiple regression analysis were performed with IBM SPSS Statistics 27.

## 3. Results

### 3.1. General Characteristics of Survey Subjects

The socio-demographic characteristics of the survey subjects, the current status of disease and drug use, the use of smart phone, and the future home for retirement were investigated in Table 2. As a result of the socio-demographic characteristics of the survey subject, the ages were 51 (48.6%) in their 40s and 54 (51.4%) in their 60s. Gender was 57 males (54.3%) and 48 females (45.7%) and occupations were office employees (14.3%), professionals (12.4%), self-employed (23.8%), government employees (12.4%), housewife (23.8%), and retired (13.3%). Depending on age, office employees in their 40s are the most common with 12 (23.6%) and retirees were 1 (1.9%), whereas for those in their 60s, 15 housewives (27.8%), self-employed 14 (25.9%), retirees 13 (24.1%). The number of family members living together was the most at 33.0%, with 5 and above people.

**Table 2.** Socio-demographic Characteristics of Survey Subjects.

| Age Group | | 40s (%) | 60s (%) | Total (%) |
|---|---|---|---|---|
| Gender | Male | 26 (51.0) | 31 (57.4) | 57 (54.3) |
| | Female | 25 (49.0) | 23 (42.6) | 48 (45.7) |
| Occupation | Office | 12 (23.6) | 3 (5.6) | 15 (14.3) |
| | Professional | 11 (21.6) | 2 (3.7) | 13 (12.4) |
| | Self-Employed | 11 (21.6) | 14 (25.9) | 25 (23.8) |
| | Government | 6 (11.7) | 7 (12.9) | 13 (12.4) |
| | Housewife | 10 (19.6) | 15 (27.8) | 25 (23.8) |
| | Retired | 1 (1.9) | 13 (24.1) | 14 (13.3) |
| Family | 1 | 0 (0) | 2 (3.7) | 2 (1.9) |
| | 2 | 7 (13.7) | 16 (29.6) | 23 (21.9) |
| | 3 | 6 (11.8) | 17 (31.5) | 23 (21.9) |
| | 4 | 11 (21.6) | 13 (24.1) | 24 (22.9) |
| | 5 and Above | 27 (52.9) | 6 (11.1) | 33 (31.4) |
| Monthly Living Expense | Under 7000 AED | 1 (1.9) | 5 (9.3) | 6 (5.7) |
| | 7000 AED | 5 (9.8) | 21 (38.9) | 26 (24.8) |
| | 10,000 AED | 8 (15.7) | 14 (25.9) | 22 (21.0) |
| | 13,000 AED | 13 (25.6) | 4 (7.4) | 17 (16.2) |
| | 17,000 AED | 12 (23.5) | 6 (11.1) | 18 (17.1) |
| | Above 20,000 AED | 12 (23.5) | 4 (7.4) | 16 (15.2) |
| Total | | 51 (100) | 54 (100) | 105 (100) |

Table 3 shows the results of the investigation of current diseases and medication use. Of the survey subjects, 33 (31.4%) responded that there was no disease. In the case of disease, diabetes was the most common with 34 (32.4%), followed by high blood pressure with 17 (16.2%). As a result of investigation of medication use, 35.2% of cases did not take any medication, 50.5% took chronic disease-related medications, and 14.3% took multivitamins or health supplements. As a result of verifying whether there is a difference in disease and medication use according to age, there was a statistically significant difference. Those in their 40s were more likely to have no disease (41.2%), and those in their 60s were more likely to take medication (64.8%). Those in their 40s (19.6%) were found to take more multivitamins or health supplements than those in their 60s (9.3%).

**Table 3.** Chronic Diseases and Medication by Age Group (* $p < 0.01$).

| Age Group | | 40s (%) | 60s (%) | Total (%) |
|---|---|---|---|---|
| Chronic Diseases | High blood pressure | 6 (11.7) | 11 (20.4) | 17 (16.2) |
| | Heart disease | 1 (2.0) | 3 (5.6) | 4 (3.8) |
| | Diabetes | 15 (29.4) | 19 (35.1) | 34 (32.4) |
| | Stomach disorder | 2 (3.9) | 2 (3.7) | 4 (3.8) |
| | Arthritis | 5 (9.8) | 6 (11.1) | 11 (10.5) |
| | Etc. | 1 (2.0) | 1 (1.9) | 2 (1.9) |
| | No chronic disease | 21 (41.2) | 12 (22.2) | 33 (31.4) |
| | $x^2 = 15.104$ * | | | |
| Medication | On medication | 18 (35.3) | 35 (64.8) | 53 (50.5) |
| | Multivitamin or supplements | 10 (19.6) | 5 (9.3) | 15 (14.3) |
| | No medication | 23 (45.1) | 14 (25.9) | 37 (35.2) |
| | $x^2 = 8.324$ * | | | |
| Total | | 51 (100) | 54 (100) | 105 (100) |

A questionnaire was developed based on the classification of smartphone use, and the degree of smartphone usage was investigated in relation to technology acceptance. For 12 questions such as daily life help, news, browser search, music, video, economy, shopping, transportation, games, entertainment, health, and social media, participants responded on a five-point Likert scale ranging from one point, 'never' to five points, 'always'. As a result of conducting the independent sample *t*-test, there were statistically significant differences according to age. For those in their 40s, most of the questions showed a high score of four points, while in the 60s, most of the questions showed two and one points (Table 4). In the case of those in their 40s, 'I search for something in Google (4.93 points)', 'I use smartphone's clock, alarm, weather, memo, or calculator (4.91 points)', 'I read news on the smartphone (4.81 points)' showed very high scores. On the other hand, 'I play games when I get bored (2.57 points)', showed the lowest score. For those in their 60s, 'I use smartphone's clock, alarm, weather, memo, or calculator' was the highest with 3.23 points, while 'I play games when I get bored' (1.54 points) and 'I communicate with my friends via Instagram and Facebook' (1.63 points) were the lowest.

**Table 4.** Smartphone Usage by Age Group (* $p < 0.01$).

| Smart Phone Activity | | 40s | 60s | *t*-Value |
|---|---|---|---|---|
| Daily Activity | I use smartphone's clock, alarm, weather, memo, or calculator, etc. | 4.91 | 3.23 | 7.07 * |
| News | I read news on the smartphone. | 4.81 | 2.28 | 13.07 * |
| Google | I search for something in Google. | 4.93 | 2.65 | 10.36 * |
| Music | I listen to music via media player. | 4.21 | 2.17 | 8.11 * |
| Youtube | I watch video on youtube.com | 4.24 | 2.04 | 9.12 * |
| Online Banking | I use online banking application. | 4.04 | 2.38 | 5.88 * |
| Email | I check work-related emails and MS documents. | 4.06 | 1.82 | 12.62 * |
| Shopping | I buy things from online shopping sites like Amazon.ae. | 4.14 | 1.88 | 8.86 * |
| Careem | I use Careem application for taxi or bike. | 4.48 | 2.21 | 10.18 * |
| Game | I play games when I get bored. | 2.57 | 1.54 | 3.76 * |
| Health | I check vital signs and exercise information from smartphone | 3.24 | 1.69 | 6.04 * |
| Social Media | I communicate with my friends via Instagram and Facebook. | 3.08 | 1.63 | 4.18 * |

By summing the scores of 12 questions related to smartphone usage, the user level was classified into three. Thus, 1 to 25 points were classified as low level users (29.6%), 26 to 45 points as medium level users (34.2%), and 46 to 60 points as high level users (36.2%). There was a statistically significant difference in the level of smartphone usage according to age, and the results are shown in Figure 1. Low level users are mainly in their 60s and high level users are mainly in their 40s.

### 3.2. Smart Home Technology Preferences

The survey subjects' need for smart home technology and sensors, and their preferred locations at home were explored. The correlation with age, gender, income, and smartphone usage was analyzed and as a result, there were significant differences according to age, gender, and level of smartphone use. It was found that the technology that the survey subjects considered most needed was environmental control technology (3.72 points), and health and biometric monitoring technology (3.58 points) (Table 5). As a result of independent sample *t*-testing between those in their 40s and 60s, in the case of those in their 60s, they preferred automatic fall detection and lifestyle monitoring technology compared to those in their 40s. For those in their 40s, it was found that they prefer environmental control, health and biometric monitoring, and video conference technology.

**Table 5.** Smart Home Technology Preference by Age Group (* $p < 0.05$, ** $p < 0.01$).

| Smart Home Technology | Total Mean | 40s ($n$ = 51) Mean | 60s ($n$ = 54) Mean | *t*-Value |
|---|---|---|---|---|
| Personal Emergency System | 2.98 | 2.87 | 3.10 | Not Significant |
| Automatic Fall Detection | 2.88 | 2.61 | 3.13 | −2.09 * |
| Lifestyle Monitoring | 2.77 | 2.48 | 3.04 | −2.18 * |
| Daily Life Assistance | 2.82 | 2.82 | 2.84 | Not Significant |
| Environmental Control | 3.71 | 4.48 | 2.99 | 6.92 ** |
| Health and Biometric Monitoring | 3.58 | 4.13 | 3.04 | 4.29 ** |
| Video Conferencing | 3.37 | 4.04 | 2.71 | 6.12 ** |

There were also significant differences according to gender (Table 6). It was found that males prefer lifestyle monitoring technology and health and biometric monitoring technology to females.

**Table 6.** Smart Home Technology Preference by Gender.

| Smart Home Technology | Male ($n$ = 57) Mean | Female ($n$ = 48) Mean | *t*-Value |
|---|---|---|---|
| Lifestyle Monitoring Technology | 3.07 | 2.51 | 2.14 * |
| Health and Biometric Monitoring Technology | 3.88 | 3.30 | 2.28 * |

* $p < 0.05$.

As a result of one-way variance analysis to understand the technology preference according to the degree of use of smartphones, there was a significant difference in part (Figure 2). In the case of environmental control, health and biometric monitoring, and video conference, it was found that the preference for medium-level users and high-level users increased over low level users.

As a result of the analysis, the most preferred types of sensors were motion detecting sensors (3.35 points), switches (3.31 points), and voice recognition sensors (3.25 points) in this order (Table 7). As a result of the correlation analysis, there were statistically significant differences in age, gender, smartphone usage, and some items excluding income. Compared to those in their 60s, those in their 40s prefer voice recognition sensors (4.00 points) and motion sensors (3.95 points).

As a result of analyzing the preference of sensor types according to gender, there was a significant difference (Table 8). Females had higher preference for switches (3.49 points) than males, and males had higher preference for human attachment sensors (2.55 points) than females.

**Table 7.** Smart Home Sensor Preference by Age Group (** $p < 0.001$).

| Smart Home Sensors | Total Mean | 40s ($n = 51$) Mean | 60s ($n = 54$) Mean | *t*-Value |
|---|---|---|---|---|
| Switches | 3.31 | 3.30 | 3.32 | Not Significant |
| Motion Detecting Sensors | 3.35 | 3.95 | 2.78 | 5.90 ** |
| Video Cameras | 2.40 | 2.48 | 2.34 | Not Significant |
| Wearable Sensors | 2.31 | 2.36 | 2.24 | Not Significant |
| Voice Recognition Sensors | 3.25 | 4.00 | 2.54 | 6.76 ** |

**Table 8.** Smart Home Sensors Preference by Gender.

| Smart Home Sensors | Male ($n = 57$) Mean | Female ($n = 48$) Mean | *t*-Value |
|---|---|---|---|
| Switches | 3.09 | 3.49 | −2.55 * |
| Wearable Sensors | 2.55 | 2.09 | 2.16 * |

* $p < 0.05$.

As a result of analyzing the preference of sensor types according to the degree of use of smartphones, there was a significant difference in part (Figure 3). High level users showed higher preference than low level users in switch (3.62 points), motion sensor (4.0 points), video camera (2.75 points), and voice recognition sensor (4.01 points).

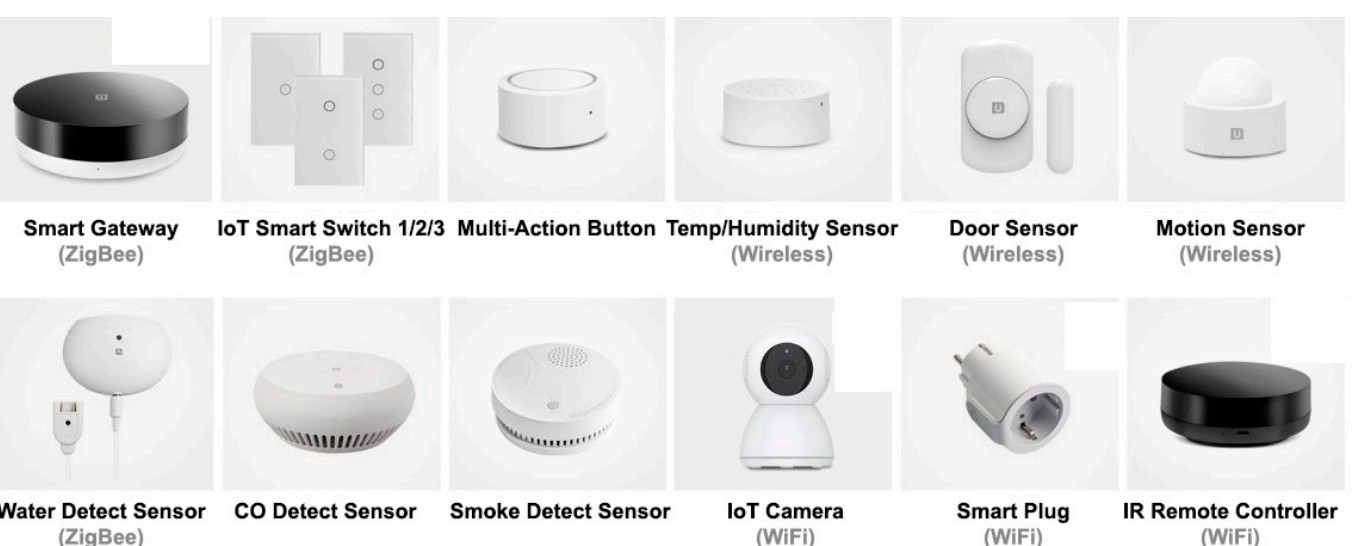

**Figure 3.** Example of Smart Devices and Sensors for Conventional Home.

Table 9 shows the results of surveying the subjects of various technologies and where sensors are appropriately located in the house. In the case of automatic fall detection, the living room (39.0%) was the most preferred, followed by the bedroom (25.7%). Lifestyle monitoring was also preferred in the order of living room (44.7%) and bedroom (18.1%). It was found that the living room was also the most preferred for installing systems such as daily life assistance (36.2%), environmental control (50.5%), health and biometric monitoring (49.5%), and video conferencing (82.9%). In the case of sensors, the switch is most preferred at the entrance (36.2%), and motion detecting sensors (42.9%), video

cameras (56.2%), and voice recognition (50.5%) sensors are preferred to be installed in the living room.

**Table 9.** Location of Smart Home Technology and Sensors.

| Category | | Entrance | Living Room | Bedroom | Kitchen | Dining Room | Bathroom | Balcony |
|---|---|---|---|---|---|---|---|---|
| SHT | Automatic Fall Detection | 4 (3.8) | 41 (39.0) | 27 (25.7) | 8 (7.6) | 2 (1.9) | 18 (17.1) | 5 (4.7) |
| | Lifestyle Monitoring | 6 (5.7) | 47 (44.7) | 19 (18.1) | 13 (12.4) | 5 (4.76) | 11 (10.5) | 4 (3.8) |
| | Daily Life Assistance | 0 (0.0) | 38 (36.2) | 18 (17.1) | 21 (20.0) | 18 (17.1) | 10 (9.5) | 0 (0.0) |
| | Environmental Control | 13 (12.4) | 53 (50.5) | 19 (18.1) | 15 (14.3) | 5 (4.8) | 0 (0.0) | 0 (0.0) |
| | Health/Biometric Monitor | 0 (0.0) | 52 (49.5) | 39 (37.1) | 0 (0.0) | 2 (1.9) | 12 (11.4) | 0 (0.0) |
| | Video Conferencing | 0 (0.0) | 87 (82.9) | 14 (13.3) | 0 (0.0) | 2 (1.9) | 2 (1.9) | 0 (0.0) |
| Sensors | Switches | 38 (36.2) | 32 (30.5) | 11 (10.4) | 8 (7.6) | 3 (2.8) | 8 (7.6) | 5 (4.8) |
| | Motion Detecting Sensors | 26 (24.8) | 45 (42.9) | 15 (14.3) | 4 (3.8) | 1 (0.9) | 10 (9.5) | 4 (3.8) |
| | Video Cameras | 11 (10.5) | 59 (56.2) | 13 (12.4) | 12 (11.4) | 2 (1.9) | 2 (1.9) | 6 (5.7) |
| | Voice Recognition Sensors | 9 (8.6) | 53 (50.5) | 15 (14.3) | 12 (11.4) | 4 (3.8) | 9 (8.6) | 3 (2.8) |

### 3.3. Smart Home Technology Acceptance and Correlation between Variables

In this study, based on the literature review, the acceptance factors of smart home technology were derived, and Table 10 shows the results of surveying users' opinions on 16 questions of seven categories on a five-point Likert scale. The category related to the facilitating conditions of smart home technology by the survey subjects FC1 (4.26), and FC2 (4.14) are high. Among the category to the expected performance of smart home technology, EP1 is 4.09 points and EP2 is 4.06 points, which are high. On the other hand, the beliefs related to the security of smart home technology are very low at 2.34 points for PS1 and 2.60 points for PS2. As a result of the internal reliability analysis of the questionnaires, the Cronbach's α value ranged from 0.738 to 0.961, exceeding the threshold of 0.700.

**Table 10.** Smart Home Technology Acceptance.

| Category | Contents | Mean | Cronbach's α |
|---|---|---|---|
| Expected Performance | EP1. | 4.09 | 0.936 |
| | EP2. | 4.06 | |
| | EP3. | 3.84 | |
| Expected Effort | EE1. | 3.46 | 0.961 |
| | EE2. | 3.31 | |
| Social Impact | SI1. | 3.58 | 0.924 |
| | SI2. | 3.63 | |
| Facilitating Conditions | FC1. | 4.26 | 0.788 |
| | FC2. | 4.14 | |
| Technology Anxiety | AT1. | 3.44 | 0.856 |
| | AT2. | 3.23 | |
| Perceived Security | PS1. | 2.34 | 0.885 |
| | PS2. | 2.60 | |
| Intention to Use | IU1. | 3.62 | 0.738 |
| | IU2. | 3.32 | |
| | IU3. | 2.75 | |

There were statistically significant differences in opinions of users on the smart home technology acceptance according to the age (Table 11) and smartphone usage level. There was no difference between those in their 40s and 60s for facilitating conditions for smart home technology, but in all other categories, those in their 40s scored higher than those in their 60s. Among the category related to Expected Performance, those in their 40s have high scores on EP1 (4.54) and EP2 (4.58). On the other hand, those in their 60s showed the highest score of 4.05 in AT2 regarding anxiety about technology.

Figure 4 shows the different results on the smart home technology acceptance according to the degree of smartphone usage level. In all categories, the score was higher as the level of smartphone usage went from low level to high level. Regarding Intention to Use, high level users scored higher than four points in IU1 and IU2.

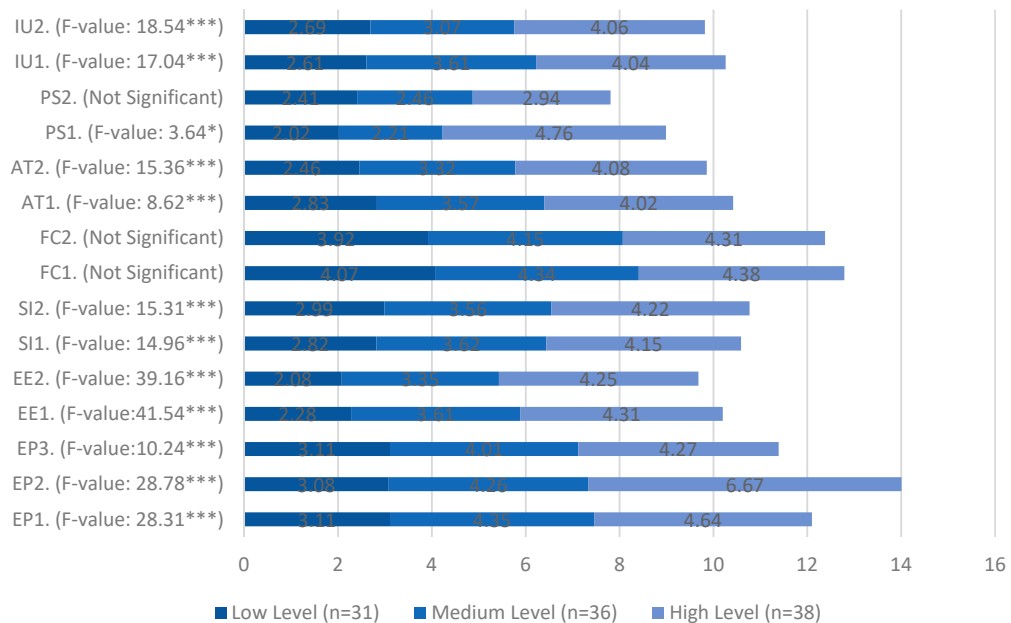

**Figure 4.** Smart Home Technology Acceptance by User Level (* $p < 0.05$, *** $p < 0.001$).

**Table 11.** Smart Home Technology Acceptance by Age Group.

| Category | | 40s ($n = 51$) Mean | 60s ($n = 54$) Mean | $t$-Value |
|---|---|---|---|---|
| Expected Performance | EP1. | 4.53 | 3.67 | 4.43 *** |
| | EP2. | 4.56 | 3.58 | 5.18 *** |
| | EP3. | 4.18 | 3.52 | 2.93 ** |
| Expected Effort | EE1. | 4.11 | 2.86 | 5.97 *** |
| | EE2. | 3.99 | 2.65 | 5.88 *** |
| Social Impact | SI1. | 3.99 | 3.19 | 3.85 *** |
| | SI2. | 4.02 | 3.26 | 3.96 *** |
| Facilitating Conditions | FC1. | 4.32 | 4.22 | Not Significant |
| | FC2. | 4.28 | 4.00 | Not Significant |
| Technology Anxiety | AT1. | 2.96 | 3.89 | −4.00 *** |
| | AT2. | 2.38 | 4.04 | −7.91 *** |
| Perceived Security | PS1. | 2.08 | 2.58 | −2.09 * |
| | PS2. | 2.34 | 2.86 | −2.21 * |
| Intention to Use | IU1. | 4.32 | 2.97 | 7.38 *** |
| | IU2. | 4.01 | 2.67 | 7.12 *** |
| | IU3. | 3.40 | 2.13 | 6.39 *** |

* $p < 0.05$, ** $p < 0.01$, *** $p < 0.001$.

## 4. Discussion

Through this research, the age and computer ability of the users are the most important variables for smart housing technology acceptance. Technology anxiety has a negative impact on users' intentions to use smart home technology, while expected performance, social impact, and facilitating conditions have a positive effect. However, in order to plan smart homes for UAE users in the future, follow-up research should be conducted on how these influencing factors are actually applied spatially, and specific solutions to negative factors to increase the use of technology. In addition, the survey in this research is limited to the apartment building type, but in future studies, research should be conducted on various building types, such as villas.

## 5. Conclusions

This study analyzed not only users' SHT needs and preferences, but also users' opinions and intentions. In addition, it was attempted to clarify whether there is a difference in preference and acceptance of smart home technologies based on different factors. The conclusions made on the basis of the analysis results on the needs and preferences of SHT and technology acceptance are as follows.

First, as a result of analyzing the characteristics of the subjects, 67.0% of the subjects have chronic diseases such as high blood pressure, heart disease, diabetes, or arthritis. Chronic diseases can be treated by improving overall lifestyles such as eating habits and exercise rather than short-term treatment at a hospital. The direction of smart home development should support daily life and health management so that users can lead a healthy life for a long time in their home.

Second, as a result of the analysis of preference for SHT, the technologies that users need most are environmental control technology and health and biometric monitoring technology. In addition, the technology need for those in their 60s, who are recognized as actually in need of SHT, is significantly lower than that for those in their 40s. The greater the technical familiarity, the higher the need for technology. Those in their 60s have greater fear of unfamiliar new technologies than those in their 40s due to lack of understanding. In addition, in terms of sensor preference, motion sensors and voice recognition sensors were found to be the most preferred. On the other hand, the preference for video cameras was very low. It is interpreted that this is because video cameras are thought to invade their privacy by exposing their private life.

Third, as a result of the analysis on the SHT acceptance, the survey subjects showed high expectation that the technology would improve the quality of their life and be useful. However, the belief that SHT would be safe was low. The help for the initial environment setup for SHT and system failure is very strong. In addition, anxiety about technology was the factor that most influenced the intention to accept SHT. It can be seen that the greater the anxiety about technology, the weaker the willingness to accept the technology.

Fourth, when applying SHT, a housing plan that considers the characteristics of various users is required. In this study, age and computer technology affinity were the most influential variables, and accordingly, there were differences in technology preference and acceptance. Since, currently, those in their 40s are the first digital generation with the internet, they grew up with mobile phones, social media, and digital environments. This generation will have a higher interest and willingness in smart health care. By 2030, when people in their 40s turn 65, the population over 65 in the United Arab Emirates is expected to increase to 24.3%. Therefore, a SHT plan customized to UAE users should take this into consideration.

This study would guide companies and researchers developing smart home and smart health products to develop systems that are better suited to users' preferences.

**Author Contributions:** All authors contributed significantly to this study. M.A., C.J. and J.A. identified and secured the example buildings used in the study. The data acquisition system and installations of sensors were designed and installed by M.A., C.J. and J.A.; Data collection was conducted by M.A. and A.H.C.; Data analysis was performed by C.J. and M.A.; The manuscript was compiled by C.J. and reviewed by M.A., J.A. and A.H.C. All authors have read and agreed to the published version of the manuscript.

**Funding:** This research received no external funding.

**Institutional Review Board Statement:** The study was conducted according to the guidelines of Ajman University Research Ethics Committee.

**Informed Consent Statement:** Informed consent was obtained from all subjects involved in the study.

**Data Availability Statement:** New data were created or analyzed in this study. Data will be shared upon request and consideration of the authors.

**Acknowledgments:** The authors would like to express their gratitude to Ajman University for the generous support for APC.

**Conflicts of Interest:** The authors declare no conflict of interest.

## Appendix A

**1. Could you describe yourself?**

| 1. Gender | Male ○ | Female ○ |
| --- | --- | --- |
| 2. Occupation | Office ○ | |
| | Professional ○ | |
| | Self-Employed ○ | |
| | Government ○ | |
| | Housewife ○ | |
| | Retired ○ | |
| 3. Number of Family | 1 ○ | |
| | 2 ○ | |
| | 3 ○ | |
| | 4 ○ | |
| | 5 & Above ○ | |
| 4. Monthly Income | Under 7000 AED ○ | |
| | 7000 AED ○ | |
| | 10,000 AED ○ | |
| | 13,000 AED ○ | |
| | 17,000 AED ○ | |
| | <20,000 AED ○ | |

**Figure A1.** *Cont.*

**2. Do you have chronic diseases or medication?**

| 1. Chronic Diseases | | |
|---|---|---|
| | High blood pressure | O |
| | Heart disease | O |
| | Diabetes | O |
| | Stomach disorder | O |
| | Arthritis | O |
| | Etc. | O |
| | No chronic disease | O |
| 2. Medication | On medication | O |
| | Multivitamin or supplements | O |
| | No medication | O |

**3. How do you use smartphone? (Please answer all the questions)**

| Questions | Never | Rarely | Sometimes | Often | Always |
|---|---|---|---|---|---|
| 1. I use smartphone's clock, alarm, weather, memo, or calculator, etc. | O | O | O | O | O |
| 2. I read news on the smartphone. | O | O | O | O | O |
| 3. I search for something in Google. | O | O | O | O | O |
| 4. I listen to music via media player. | O | O | O | O | O |
| 5. I watch video on youtube.com. | O | O | O | O | O |
| 6. I use online banking application. | O | O | O | O | O |
| 7. I check work-related emails and MS documents. | O | O | O | O | O |
| 8. I buy things from online shopping sites like Amazon.ae. | O | O | O | O | O |
| 9. I use Careem application for taxi or bike. | O | O | O | O | O |
| 10. I play games when I get bored. | O | O | O | O | O |
| 11. I check vital signs and exercise information from smartphone. | O | O | O | O | O |
| 12. I communicate with my friends via Instagram and Facebook. | O | O | O | O | O |

**4. Which "smart home technology" do you want to have?**

| | |
|---|---|
| 1. Automatic | O |
| 2. Fall Detection | O |
| 3. Lifestyle Monitoring | O |
| 4. Daily Life Assistance | O |
| 5. Environmental Control | O |
| 6. Health/Biometric Monitor | O |
| 7. Video Conferencing | O |

**5. Which "sensors" do you want to have?**

| | |
|---|---|
| 1. Switches | O |
| 2. Motion Detecting Sensors | O |
| 3. Video Cameras | O |
| 4. Voice Recognition Sensors | O |

**Figure A1.** *Cont.*

**6. Which location do you want to install "smart home technology" and "sensors"?**

| Questions | | Entrance | Living Room | Bedroom | Kitchen | Dining Room | Bathroom | Balcony |
|---|---|---|---|---|---|---|---|---|
| Smart Home Technology | Automatic Fall Detection | ○ | ○ | ○ | ○ | ○ | ○ | ○ |
| | Lifestyle Monitoring | ○ | ○ | ○ | ○ | ○ | ○ | ○ |
| | Daily Life Assistance | ○ | ○ | ○ | ○ | ○ | ○ | ○ |
| | Environmental Control | ○ | ○ | ○ | ○ | ○ | ○ | ○ |
| | Health/Biometric Monitor | ○ | ○ | ○ | ○ | ○ | ○ | ○ |
| | Video Conferencing | ○ | ○ | ○ | ○ | ○ | ○ | ○ |
| Sensors | Switches | ○ | ○ | ○ | ○ | ○ | ○ | ○ |
| | Motion Detecting Sensors | ○ | ○ | ○ | ○ | ○ | ○ | ○ |
| | Video Cameras | ○ | ○ | ○ | ○ | ○ | ○ | ○ |
| | Voice Recognition Sensors | ○ | ○ | ○ | ○ | ○ | ○ | ○ |

**Figure A1.** Smart Home Technology Acceptance and Preference Survey.

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
