# Peer review of "Analysis of Smart Home Technology Acceptance and Preference for Elderly in Dubai, UAE"

_designs, 2021_

Round 1

Reviewer 1 Report

The authors present an interesting topic titled “Smart Home Technology Acceptance and Preference for United Arab Emirates Elderly Users”. A complete, clear survey is conducted and presented here. However, this paper does not contain any in-depth analysis or elaboration of the electrical appliances used to make a conventional home into a smart home. The consumer's dependency on smart home components can play a vital role in deciding to use the smart home facilities. The authors consider only a few sensors as part of the smart home and are presented based on the consumer views/interviews. The results and the survey process are done smoothly, English and the presentations are good.  I am not sure the scope of this manuscript matches the Journals scopes, as no mathematical/simulation/hardware/physical model is designed here. The manuscript shouldn’t be accepted in its current form.

  1. The word “Survey, analysis” can be added to the title to catch the reader attention.
  2. The abstract is well written; however, more quantitative data of results can be added.
  3. More recent work on dependency on the electrical appliances of different ages users, smart home-user interaction can be added to the Introduction sections.
  4. Line 109: please check if the full term is used earlier to write the acronym “TAM” here.
  5. Table 1: Please use abbreviations as Expected Performance (EP) in all the categories.
  6. Line 155: Please check (starred) if there are any double space “First, what types of smart home technologies do UAE elderly***users prefer”
  7. Line 181: Please elaborates on the rest of the 5 questionaries that are not collected.
  8. Please use proper citation of Line 366-368. You can shift these lines to the introduction section and add the proper citation.
  9. References are enough and recent. However, please remove Line 39-400 from the reference sections.
  10. Please add any comparative table or data of this type of survey work to better understand for the reader. 

Author Response

Dear respecful reviewer,

Please kindly check the attached PDF for our revision according to your recommendation.

Thank you very much.

Reviewer 2 Report

The manuscript provides useful insight into the acceptance of smart home technologies for elderly users in the UAE which will be beneficial for businesses and researchers in developing assistance systems within the smart home domain for users in the UAE.

One suggestion would be to include the questionnaire in the appendix of the paper to help other researchers in the same study domain.

Detailed comments:
The manuscript describes a study of the acceptance of smart home technology related to health applications in elderly residents in the UAE.
This topic is an important area for the deployment and commercialization of smart home devices relating to health as well as quality of life improvement. The authors address three main questions regarding smart home technologies for the UAE, the type of technology and its location preference, factors affecting the acceptance of smart home technologies, and is there a difference in acceptance/preference
with age. The experiment is well structured and looks into important details of smart home technology for health design such as preference for the sensors used, the kind of monitoring performed, location, type of usage while also considering social and usage aspects of users anxiety, security, and privacy, as well as the effort, expected. This information is very useful in that it provides insight into the low adoption of smart home/
smart health technologies among the elderly. The results presented here would guide companies and researchers developing smart home and smart health products to develop systems that are performed more to a user's satisfaction. One thing that would help add quality to the research would be to provide the questionnaire as an appendix with the manuscript.

Author Response

(The authors gave the same response as above.)

Round 2

Reviewer 1 Report

It seems a nice improvement is done. Please follow the points is listed below:
1. Please consider the word "The" or "An" is suitable before the word "Analysis", otherwise you can start with Analysis only. 
2. Please update the Figure sequence as it is not in the right sequence. 
3. Please cite Figure 1, if you collected from a source, also consider removing the word "Smart Home*" right below.
4. Please check with Editor whether you can cite in the conclusion or not, please avoid doing several paragraphs in the conclusion.  

Author Response

Revision for Reviewer 1 (2nd Round)

  1. Please consider the word "The" or "An" is suitable before the word "Analysis", otherwise you can start with Analysis only. 

Line 2-3

“The” is removed according to reviewer’s suggestion.

Analysis of Smart Home Technology Acceptance and Preference for Elderly in Dubai, UAE

  1. Please update the Figure sequence as it is not in the right sequence. 

The “Figure sequences” is updated.

Line 80. Figure 1. Age 50 and older who have SHT in UAE (Source: Khaleej Times, 2017)

Line 125. Figure 2. Smart Home with Integrated Smart Devices

Line 163. Figure 3. Example of Smart Devices and Sensors for Conventional Home

Line 292. Figure 4. Smartphone Usage Level by Age Group (x2=58.342*, * p<.001)

Line 317. Figure 5. Smart Home Technology Preference by User Level (* p<.01, ** p<.001)

Line 337. Figure 6. Smart Home Sensors Preference by User Level (* p<.05, ** p<.01, *** p<.001)

Line 375. Figure 7. Smart Home Technology Acceptance by User Level (* p<.05, ** p<.01, *** p<.001)

  1. Please cite Figure 1, if you collected from a source, also consider removing the word "Smart Home*" right below.

Dear respectful reviewer, I designed it with Adobe Illustrator, no need to cite. 

Line 124.

"Smart Home*" is removed and newly inserted.

  1. Please check with Editor whether you can cite in the conclusion or not, please avoid doing several paragraphs in the conclusion. 

Line 417.

Citation is removed and the sentence is rewritten.

Since current 40s are the first digital generation with the Internet, they grew up with mobile phone, social media, and digital environments. This generation will have a higher interest and willingness in smart health care.
